# Species-Specific Analysis of Bacterial Vaginosis-Associated Bacteria

John Osei Sekyere,[a] Ayodeji B. Oyenihi,[a] Jason Trama,[a] Martin E. Adelson[a]

[a]Medical Diagnostic Laboratories, Genesis Biotechnology Group, Hamilton Township, New Jersey, USA

**ABSTRACT** Vaginal dysbiosis in women reduces the abundance of *Lactobacillus* species and increases that of anaerobic fastidious bacteria. This dysbiotic condition in the vagina, called bacterial vaginosis (BV), can be symptomatic with odorous vaginal discharges or asymptomatic and affects a third of women of reproductive age. Three unclassified bacterial species designated BV-associated bacteria 1, 2, and 3 (BVAB-1, -2, and -3) in 2005 were found to be highly preponderant in the vagina of females with BV. Here, we used sequence homology and phylogenetics analyses to identify the actual species of BVAB-1, -2, and -3 and found BVAB-1 to be *Clostridiales* genomosp. BVAB-1, BVAB-2 to be *Oscillospiraceae* bacterium strain CHIC02, and BVAB-3 to be *Mageeibacillus indolicus*, respectively. These are anaerobic and uncultured species that can be identified only through metagenomics. Long-read sequencing of BV specimens can also enable a genomic reassembly of these species' genomes from metagenomes. Species-specific identification of these pathogens and the availability of their genomes from assembled metagenomes will advance our understanding of their biology, facilitate the design of sensitive diagnostics and drugs, and enhance the treatment of BV.

**IMPORTANCE** For many years since 2005, BVAB, an important pathogen of the female vaginal tract that is associated with BV, has been identified using PCR without knowing its actual species. Without a full genome of these pathogens, a better understanding of their pathogenicity, treatment, resistance, and diagnostics cannot be reached. In this analysis, we use the DNA of BVAB-1, -2, and -3 to determine their actual species to enhance further research into their pathogenicity, resistance, diagnosis, and treatment.

**KEYWORDS** BV, BVAB, metagenome-assembled genome, vaginal microbiome

In 2005, Fredricks et al. (1) identified unknown bacterial species that were highly abundant in women with bacterial vaginosis (BV), a vaginal infection that affects 29.9% of women aged between 14 and 49 years in the United States (2) and 20.1% (85/422) of women in Ethiopia (3). Globally, the prevalence of BV is estimated at 20 to 30% of women of reproductive age and even higher (50 to 60%) in high-risk populations such as sex workers (4).

Using broad-range PCR and 16S rRNA primers targeted at these unknown bacteria, Fredricks et al. (1) were able to detect at least 3 clusters/phylotypes of bacteria within the class *Clostridia*; these phylotypes clustered with no known bacteria (1). As the species or genera of these novel bacteria associated with BV were not known in 2005, they were named BV-associated bacteria 1, 2, and 3 (BVAB-1, -2, and -3) to represent the 3 phylotypes of these unknown bacteria (1). Indeed, BVAB-1, -2, and -3 have been strongly associated with the vaginal microbiome of women with BV (3).

With the advent of whole-genome sequencing and metagenomics, the vaginal microbiome of women with BV has been extensively described, providing an abundant repertoire of sequences of bacteria found in women with BV (5). Hence, a species-specific resolution of what constitutes BVAB can be obtained now, while it could not be obtained before. However, short-read sequencing and the inability to cultivate many of the vaginal microbes

Address correspondence to John Osei Sekyere, joseisekyere@mdlab.com.

The authors declare no conflict of interest.

axenically in the laboratory limit the possibility of knowing the species colonizing the vaginal niche (6).

Using available sequences from metagenomic studies, comparative sequence analysis, and phylogenetics, we show that BVAB belong to newly characterized bacterial species.

**Methods.** Three 16S rRNA sequences, one each for BVAB-1, BVAB-2, and BVAB-3, were downloaded from GenBank using the accession numbers AY724739 to AY724741, respectively; these sequences were obtained from sequencing of DNA from specimens isolated from patients with BV (1). Sequence length and homology analyses were undertaken using BLAST to identify taxa or species whose genomes and DNA sequences had both 99.9 to 100% length coverage and 99.9 to 100% homology to those of BVAB-1, BVAB-2, and BVAB-3. The sequences were aligned using BLAST sequence alignment and used for downstream analysis. The genomes and sequences of the identified species and unknown bacterial species were used to undertake a fast minimum evolution phylogenetic analysis with 1,000-bootstrap sampling. The phylogenetic trees were annotated using Figtree v1.4.4. The species with the closest sequence homology and phylogenetic clustering was identified as the most closely related species or strain of BVAB-1, BVAB-2, and BVAB-3.

**Findings and conclusions.** The BVAB-1 sequence aligned with 100% homology and sequence length to several uncultured and unknown bacterial DNA sequences (at least 100 sequences), including that of a novel bacterial species called *Clostridiales* genomosp. BVAB-1 and uncultured *Lachnospiraceae* bacterium (EF120366.1, GQ358451.1, and GQ358434.1) with accession number CP049781 and BioProject number PRJNA562728 (see supplemental files S1.1 and S1.2 in the supplemental material). Phylogenetic analyses of the closely related DNA sequences showed that BVAB-1 was clustering on the same branch and clade as *Clostridiales* genomosp. (subsequently named "*Candidatus* Lachnocurva vaginae" by Holm et al. [5]), which demonstrates the close evolutionary relationship between these two sequences and shows that they are of the same species (Fig. 1). *Clostridiales* genomosp. (subsequently named "*Candidatus* Lachnocurva vaginae" by Holm et al. [5]) BVAB-1 was identified by Holm et al. from a vaginal specimen (5). They assembled the genomes of the individual species from the metagenomes sequenced using PacBio Sequel II and found BVAB-1 to be *Clostridiales* genomosp., now named "*Candidatus* Lachnocurva vaginae" (5).

As well, BVAB-2 and BVAB-3 sequences aligned with unknown and uncultivable bacteria. Specifically, BVAB-2 aligned with 100% sequence length and homology to the 16S rRNA sequences of *Oscillospiraceae* bacterium, which is a novel specie within the class *Clostridia* and includes several strains (supplemental files S1.3 and S1.4). Other taxa that also aligned with BVAB-2 were uncultured *Clostridiales* bacterium (KJ868804.1), *Acetivibrio* sp., *Ruminococcaceae*, *Ercella* sp., *Clostridium* sp., and *Saccharofermentans* sp. *Oscillospiraceae* bacterium strains CHIC02, UPII 610-J, and ACE 1034E1-9 clustered closely with BVAB-2 (Fig. 2 and Fig. S1), showing their very short evolutionary distance and confirming that they are all the same or closely related species and/or strains. *Oscillospiraceae* bacterium is not cultivable, and its metagenomically assembled genome (MAG) is also not available currently.

BVAB-3 also aligned with 100% length coverage and nucleotide homology with *Mageeibacillus indolicus* (supplemental files S1.5 and S1.6). Other taxa also found to align with BVAB-3 were *Clostridiales* genomosp. Bvab-3, uncultured *Ercella* sp. and *Acetivibrio* sp., *Ruminococcaceae* bacterium, *Saccharofermentans* sp., *Ercella succinigenes*, *Clostridiales* bacterium, and *Saccharofermentans acetigenes*. Furthermore, BVAB-3 was found within the same clade and cluster as *Mageeibacillus indolicus* with a very short evolutionary distance between them (Fig. S1). This confirms the species of BVAB-3 as *Mageeibacillus indolicus*. All DNA sequences of *Mageeibacillus indolicus* are 16S sequences that were obtained from vaginal swab specimens, and this species is not cultivable.

Furthermore, BVAB-1, BVAB-2, and BVAB-3 were phylogenetically and evolutionarily distant from each other (Fig. S2), showing that they are not of the same species. Although Holm et al. (5) found *Clostridiales* genomosp. BVAB-1 or "*Candidatus* Lachnocurva vaginae" to be closely related to *Shuttleworthia satelles* (NZ_ACIP00000000), *Lachnobacterium bovis* (GCF_900107245.1), and *Lachnospiraceae* bacterium 2_1_46FAA, we did not observe these.

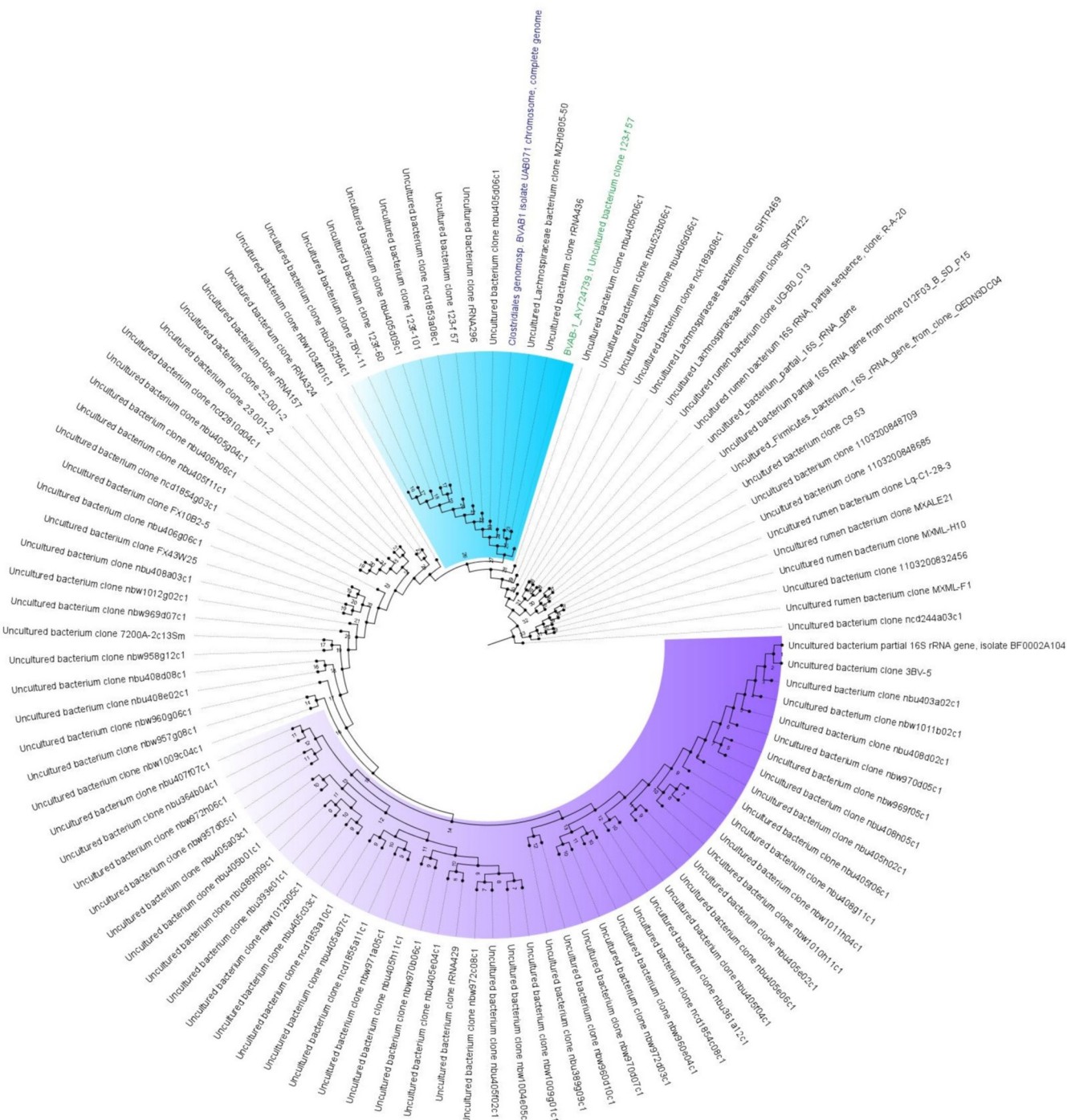

**FIG 1** Phylogenetic tree of BVAB-1 and closely related 16S rRNA/genomes. BVAB-1 is colored green (BVAB-1_AY724739.1 uncultured bacterium clone 123-f57), and the identified species that is closely related to BVAB-1, *Clostridiales* genomosp. BVAB-1 (UAB071), is colored blue. The clade bearing BVAB-1 and all other DNA sequences with a very short evolutionary distance is highlighted in blue. The clade highlighted in purple is taxa with a very short evolutionary distance but farther from BVAB-1.

This difference between our analysis and that of Holm et al. (5) could be from the source of our BVAB DNA. Whereas we used the original 16S rRNA sequences deposited at GenBank by Fredricks et al. (1), AY724739 to AY724741, Holm et al. used MAG sequences obtained from PacBio sequencing of DNA isolated from vaginal specimen of patients with BV (5).

Based on this analysis, it can be concluded that BVAB-1, BVAB-2, and BVAB-3 are either the same or closely related species and/or strains of *Clostridiales* genomosp. BVAB-1 ("*Candidatus* Lachnocurva vaginae"), *Oscillospiraceae* bacterium strain CHIC02, and

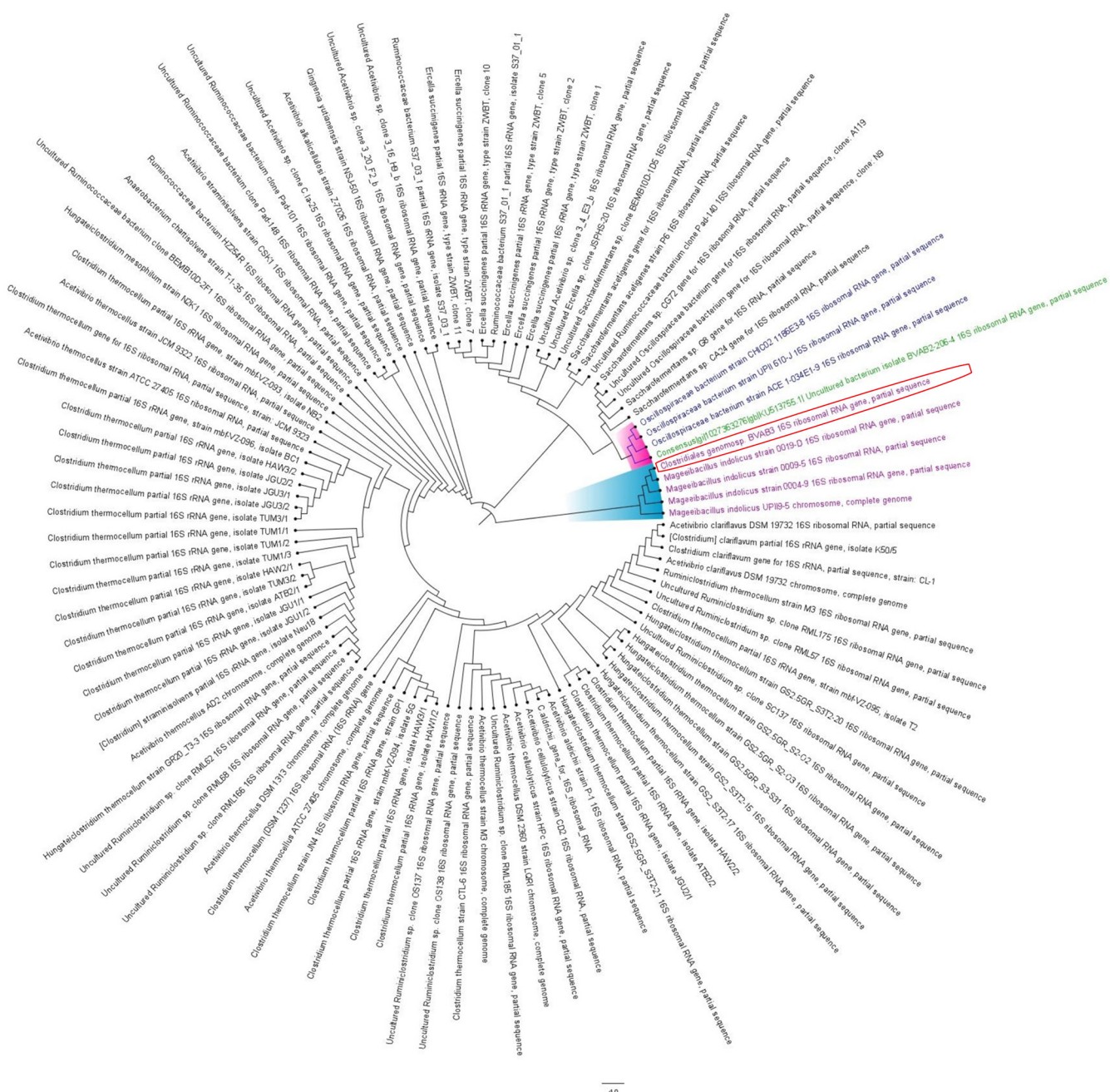

**FIG 2** Phylogenetic tree of BVAB-2, BVAB-3, and closely related 16S rRNA/genomes. BVAB-2 is colored in green, and the identified species that is closely related to BVAB-2, i.e., *Oscillospiraceae* bacterium, is colored in blue. The clade bearing BVAB-1 and all other DNA sequences with a very short evolutionary distance is highlighted in purple. BVAB-3 and all *Mageeibacillus indolicus* strains, which are of the same clade and have a very short evolutionary distance, are colored in violet while their branch is highlighted in blue. BVAB-3 is encircled in red for easy identification.

*Mageeibacillus indolicus*, respectively. Notably, these species were also obtained from metagenomes sequenced from vaginal specimens, which corroborates their strong sequence homology with BVAB-1, -2, and -3. The full genomes of *Oscillospiraceae* bacterium (BVAB-2) and *Mageeibacillus indolicus* (BVAB-3) can be obtained only from long-read sequencing using either Oxford Nanopore or PacBio's Sequel IIe sequencing of vaginal specimens from women with BV (5, 6). The genomes can then be binned from the metagenomes. This is especially the case as these bacteria cannot be cultivated axenically in the laboratory (5, 6).

Owing to the importance of BVAB identification for the diagnosis and treatment of BV in women, a definitive species analysis is very critical for the management of this common female genital disease condition. Currently, all BVAB diagnoses are done using PCR and/or

sequencing. Hence, identifying the species of these important bacteria will advance the study, understanding, and design of novel treatments and diagnostics.

**Data availability.** All data used for this work are found in the supplemental material.

## SUPPLEMENTAL MATERIAL

Supplemental material is available online only.

**SUPPLEMENTAL FILE 1**, PDF file, 0.7 MB.
**SUPPLEMENTAL FILE 2**, JPG file, 1.9 MB.
**SUPPLEMENTAL FILE 3**, JPG file, 0.4 MB.
**SUPPLEMENTAL FILE 4**, DOCX file, 0.01 MB.

## ACKNOWLEDGMENTS

This work was funded by Medical Diagnostic Laboratories, a subsidiary of Genesis Biotechnology Group.

All authors are employees of Medical Diagnostic Laboratories, which specialize in the clinical diagnosis of female infectious diseases.

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
