## [Reviewer comments · Microbiology Spectrum]

Microbiology Spectrum

Species-specific analysis of bacterial vaginosis (BV)- associated bacteria (BVAV)

John Osei Sekyere, Ayodeji Oyenihi, Jason Trama, and Martin Adelson

Corresponding Author(s): John Osei Sekyere, University of Pretoria

Review Timeline:

Submission Date:	November 16, 2022
Editorial Decision:	April 12, 2023
Revision Received:	May 11, 2023
Accepted:	June 4, 2023

Editor: Steven Frese

Reviewer(s): Disclosure of reviewer identity is with reference to reviewer comments included in decision letter(s). The following individuals involved in review of your submission have agreed to reveal their identity: Francescopaolo Antonucci (Reviewer #1)

Transaction Report:

DOI: <https://doi.org/10.1128/spectrum.04676-22>

April 12, 2023

Dr. John Osei Sekyere
University of Pretoria
Department of Dermatology
School of Medicine
Faculty of Health Sciences
Pretoria, Gauteng 0084
South Africa

Re: Spectrum04676-22 (Species-specific analysis of bacterial vaginosis (BV)-associated bacteria (BVAV))

Dear Dr. John Osei Sekyere:

I know this has taken some time, but thank you for your patience. I'd like to move forward with the reviews we do have below.

Link Not Available

Sincerely,

Steven Frese

Journals Department
Reviewer comments:

Reviewer #1 (Comments for the Author):

Reviewer' Comments to Author:

The authors determined the actual species of Bacterial-Vaginosis associated Bacteria (BVAB) -1, -2 and -3 through phylogenetic analysis of their respective DNA sequences.

From this article it can be concluded that the species of BVAB-1 is Clostridials genomosp., BVAB-2 is Oscillospiraceae bacterium, and BVAB-3 is Mageeibacillus indolicus.

I due have some questions/suggestions to clarify few aspects and about the interpretation of the results:

QUESTION 1= How many sequences have you downloaded from GenBank?

QUESTION 2= Have you assessed the number of taxa among the used libraries?

QUESTION 3= Which were the characteristics of the patients from whom these strains were isolated? Were they all with a diagnosed BV? More discussion should be written about it in the "RESULTS AND DISCUSSION" section.

LINE 18-19= Please consider to readapt this sentence. Is BVAB-1 a *O. bacterium* or *C. genomospecies*? What about *M. indolicus*?

LINE 34= Please, provide reference.

LINE 48= Please, provide reference.

LINE 52= Please, provide reference.

LINE 68-76= Have you noticed any phylogenetic similarities of BVAB-1 with *Shuttleworthia satelles*? Were there any sequences of *S. satelles* that clustered with the BVAB-1 sequences? Considering recent results on literature, this should be clarified or at least mentioned.

LINE 74= Please, provide reference and add it also into the "REFERENCES" paragraph.

LINE 76= Please, check the correct reference and report it.

LINE 93= It would be worth to add a phylogenetic tree that includes not just BVAB-1, -2 and -3 but also some of the sequences that are located nearby this cluster.

LINE 94-96= Please, consider to clarify this sentence; are they same species or strains? Are they different species? Is BVAB-2 and -3 *O. bacterium* and *M. indolicus*, respectively?

FIGURE 1= I am assuming that the Figure 1 is the figure located at page 7. Please, consider to explain better this Figure description. Is the BVAB-1 the isolate UAB071? What about the purple clade?

FIGURE 2= I would suggest to colour the BVAB-3 strain with a different colour like you did for BVAB-2; this will help the identification of it among all the branches.

FIGURE S1= Are these sequences included in the phylogenetic tree of Figure 2 as well? I would recommend to include these sequences into the Figure 2 for better view of these clusters.

RESULTS AND DISCUSSION= Please, report in this section the results and their interpretations only; it is recommended not to mention further facts, such as the current methods used for the BVAB diagnosis. I would recommend to focus the attention more on the potential novel identified species, giving further explanation of the obtained results.

Staff Comments:

Preparing Revision Guidelines

Please return the manuscript within 60 days; if you cannot complete the modification within this time period, please contact me. If you do not wish to modify the manuscript and prefer to submit it to another journal, please notify me of your decision immediately so that the manuscript may be formally withdrawn from consideration by Microbiology Spectrum.

Reviewer' Comments to Author:

The authors determined the actual species of Bacterial-Vaginosis associated Bacteria (BVAB) -1, -2 and -3 through phylogenetic analysis of their respective DNA sequences. From this article it can be concluded that the species of BVAB-1 is Clostridiales genomosp., BVAB-2 is Oscillospiraceae bacterium, and BVAB-3 is *Mageeibacillus indolicus*.

I do have some questions/suggestions to clarify few aspects and about the interpretation of the results:

QUESTION 1= How many sequences have you downloaded from GenBank?

QUESTION 2= Have you assessed the number of taxa among the used libraries?

QUESTION 3= Which were the characteristics of the patients from whom these strains were isolated? Were they all with a diagnosed BV? More discussion should be written about it in the "RESULTS AND DISCUSSION" section.

LINE 18-19= Please consider to readapt this sentence. Is BVAB-1 a *O.* bacterium or *C.* genomospecies? What about *M. indolicus*?

LINE 34= Please, provide reference.

LINE 48= Please, provide reference.

LINE 52= Please, provide reference.

LINE 68-76= Have you noticed any phylogenetic similarities of BVAB-1 with *Shuttleworthia* satellites? Were there any sequences of *S. satellites* that clustered with the BVAB-1 sequences? Considering recent results on literature, this should be clarified or at least mentioned.

LINE 74= Please, provide reference and add it also into the "REFERENCES" paragraph.

LINE 76= Please, check the correct reference and report it.

LINE 93= It would be worth to add a phylogenetic tree that includes not just BVAB-1, -2 and -3 but also some of the sequences that are located nearby this cluster.

LINE 94-96= Please, consider to clarify this sentence; are they same species or strains? Are they different species? Is BVAB-2 and -3 *O.* bacterium and *M. indolicus*, respectively?

FIGURE 1= I am assuming that the Figure 1 is the figure located at page 7. Please, consider to explain better this Figure description. Is the BVAB-1 the isolate UAB071? What about the purple clade?

FIGURE 2= I would suggest to colour the BVAB-3 strain with a different colour like you did for BVAB-2; this will help the identification of it among all the branches.

FIGURE S1= Are these sequences included in the phylogenetic tree of Figure 2 as well? I would recommend to include these sequences into the Figure 2 for better view of these clusters.

RESULTS AND DISCUSSION= Please, report in this section the results and their interpretations only; it is recommended to not mention further facts, such as the current methods used for the BVAB diagnosis. I would recommend to focus the attention more on the potential novel identified species, giving further explanation of the obtained results.

Reviewer comments	Response
QUESTION 1= How many sequences have you downloaded from GenBank?	Three 16S rRNA sequences, one each for BVAB-1, BVAB-2, and BVAB-3, were downloaded from GenBank for determining the taxa associated with them. See line 56
QUESTION 2= Have you assessed the number of taxa among the used libraries?	The taxa that aligned to the BVAB-1, BVAB-2, and BVAB-3 were mainly uncultured and undefined so it's impossible to assess the number of taxa. However, at least 100 sequences were closely aligned to the BLASTed sequences. For BVAB-1, the main taxa besides C. genomospo were Lachnospiraceae bacterium. See lines 69-71. For BVAB-2 & BVAB-3, the other taxa are detailed in lines 82-84, and 90-93.
QUESTION 3= Which were the characteristics of the patients from whom these strains were isolated? Were they all with a diagnosed BV? More discussion should be written about it in the "RESULTS AND DISCUSSION" section.	These strains were not isolated from patients, but they were identified from specimens obtained from patients with BV (bacterial vaginosis). Lines 59-60
LINE 18-19= Please consider to readapt this sentence. Is BVAB-1 a O. bacterium or C. genomospecies ? What about M. indolicus ?	Thanks. I modified these lines accordingly.
LINE 34= Please, provide reference.	Please note that line 34 already has an in-text citation: Fredericks et al. (2005)
LINE 48= Please, provide reference.	Provided in line 50
LINE 52= Please, provide reference.	Provided in line 53
LINE 68-76= Have you noticed any phylogenetic similarities of BVAB-1 with Shuttleworthia satelles? Were there any sequences of S. satelles that clustered with the BVAB-1 sequences? Considering recent results on literature, this should be clarified or at least mentioned.	Thanks for bringing this to our attention. Please see lines 101-110 for these details.
LINE 74= Please, provide reference and add it also into the "REFERENCES" paragraph.	An in-text citation of this is found in line 80. Also provided in the reference list.

LINE 76= Please, check the correct reference and report it.	Provided in line 82
LINE 93= It would be worth to add a phylogenetic tree that includes not just BVAB-1, -2 and -3 but also some of the sequences that are located nearby this cluster.	These are already shown in Figures 1, 2, and S1.
LINE 94-96= Please, consider to clarify this sentence; are they same species or strains? Are they different species? Is BVAB-2 and -3 O. bacterium and M. indolicus, respectively?	Thanks for drawing our attention to this. I added “respectively” to it to distinguish it. We used strains/species because some of them are strains while others are species. For instance, M. indolicus is species while O. bacterium CHIC02 is a strain. As these organisms are still being investigated, it’s best to use strains and species to distinguish them as the same strains can even be later found to belong to different species.
FIGURE 1= I am assuming that the Figure 1 is the figure located at page 7. Please, consider to explain better this Figure description. Is the BVAB-1 the isolate UAB071? What about the purple clade?	Yes. Figure 1 is located at page 7. Thanks for your recommendation. We elaborated the figure legend/description in lines 161-165.
FIGURE 2= I would suggest to colour the BVAB-3 strain with a different colour like you did for BVAB-2; this will help the identification of it among all the branches.	Thanks. I have encircled it in red to distinguish it, while maintaining the coloring to show that it’s closely related to M. indolicus
FIGURE S1= Are these sequences included in the phylogenetic tree of Figure 2 as well? I would recommend to include these sequences into the Figure 2 for better view of these clusters.	Yes, Figure 2 is a combination of BVAB-2 and BVAB-3 (and associated sequences), but Fig. S1 is only BVAB-3.
RESULTS AND DISCUSSION= Please, report in this section the results and their interpretations only; it is recommended not to mention further facts, such as the current methods used for the BVAB diagnosis. I would recommend to focus the attention more on the potential novel identified species, giving further explanation of the obtained results.	Thanks for the recommendation. Indeed, we only mentioned the diagnosis aspect in the concluding paragraph (line 125) as a supporting argument for the need to know the species of these unknown organisms. The mention of PCR & sequencing was therefore not the focus, but an example of the importance of this work.

June 4, 2023

Dr. John Osei Sekyere
University of Pretoria
Department of Dermatology
School of Medicine
Faculty of Health Sciences
Pretoria, Gauteng 0084
South Africa

Re: Spectrum04676-22R1 (Species-specific analysis of bacterial vaginosis (BV)-associated bacteria (BVAV))

Dear Dr. John Osei Sekyere:

Your manuscript has been accepted, and I am forwarding it to the ASM Journals Department for publication. You will be notified when your proofs are ready to be viewed.

Sincerely,

Steven Frese
Editor, Microbiology Spectrum
